# FEW-SHOT REGRESSION VIA LEARNING SPARSIFYING BASIS FUNCTIONS

## ABSTRACT

Recent few-shot learning algorithms have enabled models to quickly adapt to new tasks based on only a few training samples. Previous few-shot learning works have mainly focused on classification and reinforcement learning. In this paper, we propose a method that focuses on regression tasks. Our model is based on the idea that the degree of freedom of the unknown function can be significantly reduced if it is represented as a linear combination of a set of *sparsifying basis functions*. This enables using a few labelled samples to learn a good approximation of the entire function. We design a *Basis Function Learner* network to encode the basis functions for a task distribution, and a *Weights Generator* to generate the weight vector for a novel task. We show that our model outperforms current state of the art meta-learning methods in various regression tasks.

## 1 INTRODUCTION

Regression deals with the problem of learning a model relating a set of inputs to a set of outputs. The learned model can be thought as function $\boldsymbol{y} = F(\boldsymbol{x})$ that gives a prediction $\boldsymbol{y} \in \mathbb{R}^{d_y}$ given input $\boldsymbol{x} \in \mathbb{R}^{d_x}$ where $d_y$ and $d_x$ are dimensions of the output and input respectively. Typically, a regression model is trained on a large number of data points to be able to provide accurate predictions for new inputs. Recently, there have been a surge in popularity on *few-shot learning* methods (Vinyals et al., 2016; Koch et al., 2015; Gidaris & Komodakis, 2018). Few-shot learning methods require only a few examples from each task to be able to quickly adapt and perform well on a new task. These few-shot learning methods in essence are *learning to learn* i.e. the model learns to quickly adapt itself to new tasks rather than just learning to give the correct prediction for a particular input sample.

In this work, we propose a few shot learning model that targets few-shot regression tasks. Our model takes inspiration from the idea that the degree of freedom of $F(\boldsymbol{x})$ can be significantly reduced when it is modeled a linear combination of sparsifying basis functions. Thus, with a few samples, we can estimate $F(\boldsymbol{x})$. The two primary components of our model are (i) the Basis Function Learner network which encodes the basis functions for the distribution of tasks, and (ii) the Weights Generator network which produces the appropriate weights given a few labelled samples. We evaluate our model on the sinusoidal regression tasks and compare the performance to several meta-learning algorithms. We also evaluate our model on other regression tasks, namely the 1D heat equation tasks modeled by partial differential equations and the 2D Gaussian distribution tasks. Furthermore, we evaluate our model on image completion as a 2D regression problem on the MNIST and CelebA data-sets, using only a small subset of known pixel values. To summarize, our contributions for this paper are:

- We propose to address few shot regression by linear combination of a set of sparsifying basis functions.

- We propose to learn these (continuous) sparsifying basis functions from data. Traditionally, basis functions are hand-crafted (e.g. Fourier basis).

- We perform experiments to evaluate our approach using sinusoidal, heat equation, 2D Gaussian tasks and MNIST/CelebA image completion tasks.

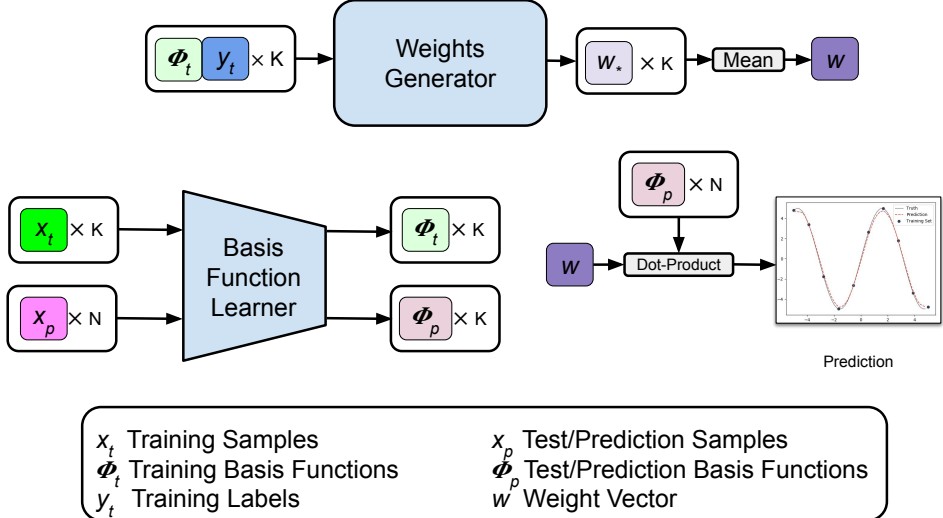

Figure 1: An overview of our model as in *meta-training*. Our system learns the basis functions $\Phi$ that can result in sparse representation for *any task* drawn from a certain task distribution. The basis functions are encoded in the Basis Function Learner network. The system produces predictions for a regression task by generating a weight vector, $w$ for a novel task, using the Weights Generator network. The prediction is obtained by taking a dot-product between the weight vector and learned basis functions.

## 2 RELATED WORK

Regression problems has long been a topic of study in the machine learning and signal processing community (Myers & Myers, 1990; Specht, 1991). Though similar to classification, regression estimates one or multiple scalar values and is usually thought of as a single task problem. A single model is trained to only perform regression on only one task. Our model instead reformulates the regression problem as a few-shot learning problem, allowing for our model to be able to perform regressions of tasks sampled from the same task distribution.

The success achieved by deep neural networks heavily relies on a large amount of data, especially labelled ones. As labelling data is time-consuming and labor-intensive, learning from limited labelled data is drawing more and more attention. A prominent approach is meta learning. Meta learning, also referred as learning to learn, aims at learning an adaptive model across different tasks. Meta learning has shown potential in style transfer (Zhang et al., 2019), visual navigation (Wortsman et al., 2018), etc. Meta learning has also been applied to few-shot learning problems, which concerns models that can learn from prior experiences to adapt to new tasks. Lake et al. (2011) proposed the one-shot classification problem and introduced the Omniglot data set as a few-shot classification data set, similar to MNIST (LeCun, 1998) for traditional classification. Since then, there has been a surge of meta learning methods striving to solve few-shot problems. Some meta learning approaches learn a similarity metric (Snell et al., 2017; Vinyals et al., 2016; Koch et al., 2015) between new test examples with few-shot training samples to make the prediction. The similarity metric used here can be Euclidean distance, cosine similarity or more expressive metric learned by relation networks (Sung et al., 2018). On the other hand, optimization-based approaches learn how to optimize the model directly. Finn et al. (2017) learned an optimal initialization of models for different tasks in the same distribution, which is able to achieve good performance by simple gradient descent. Rusu et al. (2019) learned how to perform gradient descent in the latent space to adapt the model parameters more effectively. Ravi & Larochelle (2016) employed an LSTM to learn an optimization algorithm. Generative models are also proposed to overcome the limitations resulted from few-shot setting (Zhang et al., 2018; Hariharan & Girshick, 2017; Wang et al., 2018) .

Few-shot regression tasks are used among various few-shot leaning methods (Finn et al., 2017; Rusu et al., 2019; Li et al., 2017). In most existing works, these experiment usually does not extend beyond the sinusoidal and linear regression tasks.

**A prominent family of algorithms** that tackles a similar problem as few-shot regression is Neural Processes (Garnelo et al., 2018b;a; Kim et al., 2019). Neural Processes algorithms model the distributions of the outputs of regression functions using Deep Neural Networks given pairs of input-output pairs. Similar to Variational Autoencoders (Kingma & Welling, 2013), Neural Processes employ a Bayesian approach in modelling the output distribution of regression function using an encoder-decoder architecture. Our model on the other hand employs a deterministic approach where we directly learn a set of basis functions to model the output distribution. Our model also does not produce any latent vectors but instead produces predictions via a dot product between the learned basis functions and weight vector. Our experiment results show that our model (based on sparse linear combination of basis functions) compares favorably to Neural Processes (based on conditional stochastic processes).

Our proposed sparse linear representation framework for few shot regression makes the few shot regression problem appears to be similar to another research problem called dictionary learning (DL) (Tosic & Frossard, 2011), which focuses on learning dictionaries of atoms that provide efficient representations of some class of signals. However the differences between DL and our problem are significant: Our problems are continuous rather than discrete as in DL, and we only observe a very small percentage of samples. Detailed comparison with DL is discussed in the appendix.

## 3 PROPOSED METHOD

### 3.1 PROBLEM FORMULATION

We first provide problem definition for few-shot regression. We aim at developing a model that can rapidly regress to a variety of equations and functions based on only a few training samples. We assume that each equation we would like to regress is a task $\mathcal{T}_i$ sampled from a distribution $p(\mathcal{T})$. We train our model on a set of training tasks, $\mathbb{S}_{train}$, and evaluate it on a separate set of testing tasks, $\mathbb{S}_{test}$. Unlike few-shot classification tasks, the tasks distribution $p(\mathcal{T})$ is continuous for regression task in general. Each regression task is comprised of training samples $\mathbb{D}_{train}$ and validation samples $\mathbb{D}_{val}$, for both the training set $\mathbb{S}_{train}$ and testing set $\mathbb{S}_{test}$, $\mathbb{D}_{train}$ is comprised of $K$ training samples and labels $\mathbb{D}_{train} = \{(\boldsymbol{x}_t^k, \boldsymbol{y}_t^k)|k = 1...K\}$ while $\mathbb{D}_{val}$ is comprised of $N$ samples and labels $\mathbb{D}_{val} = \{(\boldsymbol{x}_p^n, \boldsymbol{y}_p^n)|n = 1...N\}$. The goal of few-shot regression is to regress the *entire, continuous* output range of the equation given only the few points as training set.

### 3.2 FEW-SHOT REGRESSION VIA LEARNING SPARSIFYING BASIS FUNCTIONS

Here we discuss our main idea. We would like to model the unknown function $\boldsymbol{y} = F(\boldsymbol{x})$ given only $\mathbb{D}_{train} = \{(\boldsymbol{x}_t^k, \boldsymbol{y}_t^k)|k = 1...K\}$. With small $K$, e.g. $K = 10$, this is an ill-posed task, as $F(\boldsymbol{x})$ can take any form. As stated before, we assume that each function we would like to regress is a task $\mathcal{T}_i$ drawn from an *unknown* distribution $p(\mathcal{T})$.

To simplify discussion, we assume scalar input and scalar output. Our idea is to learn *sparse* representation of the unknown function $F(x)$, so that a few samples $\{(x_t^k, y_t^k)|k = 1...K\}$ can provide adequate information to approximate the entire $F(x)$. Specifically, we model the unknown function $F(x)$ as a linear combination of a set of *basis functions* $\{\phi_i(x)\}$:

$$F(x) = \sum_i w_i \phi_i(x) \tag{1}$$

Many *handcrafted* basis functions have been developed to expand $F(x)$. For example, the Maclaurin series expansion (Taylor series expansion at $x = 0$) uses $\{\phi_i(x)\} = \{1, x, x^2, x^3, ...\}$:

$$F(x) = w_0 + w_1 x + w_2 x^2 + ... \tag{2}$$

If $F(x)$ is a polynomial, (2) can be a sparse representation, i.e. only a few non-zero, significant $w_i$, and most $w_i$ are zero or *near zero*. However, if $F(x)$ is a sinusoid, it would require many terms to represent $F(x)$ adequately, e.g.:

$$\sin(x) \approx w_1 x + w_3 x^3 + w_5 x^5 + w_7 x^7 + ... + w_M x^M \tag{3}$$

In (3), $M$ is large and $M \gg K$. Given only $K$ samples $\{(x_t^k, y_t^k)|k = 1...K\}$, it is not adequate to determine $\{w_i\}$ and model the unknown function. On the other hand, if we use the Fourier basis

instead, i.e., $\{\phi_i(x)\} = \{1, \sin(x), \sin(2x), ..., \cos(x), \cos(2x), ...\}$, clearly, we can obtain a sparse representation: we can adequately approximate the sinusoid with only a few terms. Under Fourier basis, there are only a few non-zero significant weights $w_i$, and $K$ samples are sufficient to estimate the significant $w_i$ and approximate the function. **Essentially, with a sparsifying basis $\{\phi_i(x)\}$, the degree of freedom of $F(x)$ can be significantly reduced when it is modeled using (1), so that $K$ samples can well estimate $F(x)$.**

Our approach is to use the set of training tasks drawn from $p(\mathcal{T})$ to learn $\{\phi_i(x)\}$ that result in sparse representation for *any task* drawn from $p(\mathcal{T})$. The set of $\{\phi_i(x)\}$ is encoded in the **Basis Function Learner Network** that takes in $x$ and outputs $\Phi(x) = [\phi_1(x), \phi_2(x), ..., \phi_M(x)]^T$. In our framework, $\Phi(x)$ *is the same for any task drawn from $p(\mathcal{T})$*, as it encodes the set of $\{\phi_i(x)\}$ that can sparsely represent any task from $p(\mathcal{T})$. We further learn a **Weights Generator Network** to map the $K$ training samples of a novel task to a constant vector $\boldsymbol{w} = [w_1, w_2, ..., w_M]^T$. The unknown function is modeled as $\boldsymbol{w}^T \Phi(x)$.

## 3.3 MODEL ARCHITECTURE

An overview of our model is depicted in Figure 1. Given a regression task $\mathcal{T}$ with $\mathbb{D}_{train} = \{(\boldsymbol{x}_t^k, \boldsymbol{y}_t^k) | k = 1...K\}$, the model is tasked to approximate the function across the entire output. The training samples, $\boldsymbol{x}_t^k \in \mathbb{R}^{d_x}$ first passed though the **Basis Function Learner** which is represented as a network $\Phi_\theta(\boldsymbol{x})$, parameterized by trainable parameters $\theta$. The Basis Function Learner outputs set of learned basis functions in the form of a vector, $\Phi(\boldsymbol{x}) \in \mathbb{R}^{d_\phi}$, where $d_\phi$ is the number of the basis functions we would like the Basis Function Learner to learn. We represent the Basis Function Learner as a series of fully connected layers followed by a ReLU non-linearity activation function (Nair & Hinton, 2010).

The set of learned basis functions $\Phi(\boldsymbol{x})$, together with the labels $\boldsymbol{y}_t^k \in \mathbb{R}^{d_y}$ are then passed into the **Weights Generator**. The Weights Generator, represented as a network $G_\psi(\Phi(\boldsymbol{x}_t^k), \boldsymbol{y}_t^k)$, with trainable parameters $\psi$, takes the input $\Phi(\boldsymbol{x}_t^k), \boldsymbol{y}_t^k$ and outputs a weights vector $\boldsymbol{w}_k$ for each training sample of a regression task. The final weights vector, $\boldsymbol{w}$ for task $\mathcal{T}$ is then obtained by taking a mean of the $K$ weight vectors. The Weights Generator consists a series of $B$ self attention blocks following by a final fully connected layer to transform the output into the desired dimensions. We provide architecture details of Weights Generator network in the appendix.

The model is then applied to make prediction for any input $\boldsymbol{x}$. During meta-training, the validation set $\mathbb{D}_{val} = \{\boldsymbol{x}_p^n, \boldsymbol{y}_p^n | n = 1...N\}$ for a task $\mathcal{T}$ is given. The prediction is produced by taking a dot product between task-specific weights vector, $\boldsymbol{w}$ and the set of learned basis functions:

$$\boldsymbol{y}_{pred}^n = \boldsymbol{w}^T \Phi_\theta(\boldsymbol{x}_p^n) \tag{4}$$

To train our model, we design a loss function $\mathcal{L}$ that consists of three terms. The first term is a mean-squared error between the validation set labels $\boldsymbol{y}_p^n \in \mathbb{D}_{val}$ and the predicted $\boldsymbol{y}_{pred}^n$. We also add two penalty terms on the weights vector $\boldsymbol{w}$ generated for each task. The first penalty term is on the L1 norm of the generated weight vectors. This is to encourage the learned weight vectors to be sparse in order to approximate the unknown function with a few significant basis functions. The second penalty term is on the L2 norm of the generated weights vector. This is used to reduce the variance of the estimated weights as commonly used in regression (Zou & Hastie (2005)). The full loss function $\mathcal{L}$ is as follows:

$$\mathcal{L}_{\theta, \psi} = \frac{1}{N} \sum_{\boldsymbol{y}_p^n \in \mathbb{D}_{val}} (\boldsymbol{y}_p^n - \boldsymbol{y}_{pred}^n)^2 + \lambda_1 ||\boldsymbol{w}^T||_1 + \lambda_2 ||\boldsymbol{w}^T||_2 \tag{5}$$

where $\lambda_1$ and $\lambda_2$ represents the weightage of the L1 and L2 terms respectively. Note that, it turns out that our loss function for meta learning is is similar to that of the Elastic Net Regression (Zou & Hastie, 2005) with both L1 and L2 regularization terms. However, the difference is significant: Instead of focusing on a single regression task as in (Zou & Hastie, 2005), we use this loss function to learn (i) the parameter $\theta$ for the Basis Function Learner network, which encodes the sparsifying basis functions for *any task* drawn from a task distribution, and (ii) the parameter $\psi$ for the Weight Generator network, which produces the weights for *any novel task* drawn from the same task distribution.

Table 1: Mean-Squared Error results for the Sinusoidal Regression task as compared against other methods. Lower is better.

| Method | 5-shot | 10-shot | 20-shot |
|---|---|---|---|
| MAML (Finn et al., 2017) | $1.13 \pm 0.18$ | $0.77 \pm 0.11$ | $0.48 \pm 0.08$ |
| Meta-SGD (Li et al., 2017) | $0.90 \pm 0.16$ | $0.53 \pm 0.09$ | $0.31 \pm 0.05$ |
| EMAML (small)(Yoon et al., 2018) | $0.885 \pm 0.117$ | $0.615 \pm 0.091$ | $0.371 \pm 0.048$ |
| EMAML (large) | $0.783 \pm 0.101$ | $0.537 \pm 0.079$ | $0.307 \pm 0.040$ |
| BMAML (small)(Yoon et al., 2018) | $0.927 \pm 0.116$ | $0.735 \pm 0.104$ | $0.459 \pm 0.058$ |
| BMAML (large) | $0.878 \pm 0.108$ | $0.675 \pm 0.094$ | $0.442 \pm 0.055$ |
| NP (Garnelo et al., 2018b) | $0.640 \pm 0.205$ | $0.561 \pm 0.234$ | $0.421 \pm 0.088$ |
| CNP (Garnelo et al., 2018a) | $0.910 \pm 0.234$ | $0.630 \pm 0.222$ | $0.393 \pm 0.145$ |
| ANP (Kim et al., 2019) | $0.488 \pm 0.188$ | $0.216 \pm 0.082$ | $0.095 \pm 0.068$ |
| Ours (small) | $\mathbf{0.363 \pm 0.018}$ | $\mathbf{0.169 \pm 0.007}$ | $\mathbf{0.076 \pm 0.004}$ |
| Ours (large) | $\mathbf{0.199 \pm 0.010}$ | $\mathbf{0.062 \pm 0.003}$ | $\mathbf{0.027 \pm 0.002}$ |

## 4 EXPERIMENTS AND ANALYSIS

In this section we describe the experiments we ran and introduce the types of regression task used to evaluate our method. For all of our experiments, we set the learning rate to $0.001$ and use the Adam Optimizer (Kingma & Ba, 2014) as the optimization method to preform stochastic gradient decent on our model. We implement all our models using the Tensorflow (Abadi et al., 2016) library. In the following subsections, we decribe each of experiments in more detail. We include the experiments on the 1D Heat Equation and 2D Gaussian regression tasks in the appendix.

### 4.1 1D REGRESSION

For all 1D Regression tasks, the Basis Function Learner consists of two fully connected layers with 40 hidden units. For the loss function we set $\lambda_1 = 0.001$ and $\lambda_2 = 0.0001$.

**Sinusoidal Regression**. We first evaluate our model on the sinusoidal regression task which is a few-shot regression task that is widely used by other few-shot learning methods as a few-shot learning task to evaluate their methods on (Finn et al., 2017; Li et al., 2017; Rusu et al., 2019). The target function is defined as $y(x) = A sin(\omega x + b)$, where amplitude $A$, phase $b$, frequency $\omega$ are the parameters of the function. We follow the setup exactly as in (Li et al., 2017). We sample the each parameters uniformly from range $A \in [0.1, 5.0]$, $b \in [0, \pi]$ and $\omega \in [0.8, 1.2]$. We train our model on tasks of batch size 4 and 60000 iterations for 5,10 and, 20 shot cases, where each training task contains $K \in \{5, 10, 20\}$ training samples and 10 validation samples. We compare our method against recent few-shot learning methods including Meta-SGD (Li et al., 2017), MAML (Finn et al., 2017), EMAML ,BMAML (Yoon et al., 2018) and the Neural Processes family of methods including Neural Processes (Garnelo et al., 2018b) Conditional Neural Processes (Garnelo et al., 2018a) and Attentive Neural Processes (Kim et al., 2019). We use the officially released code for these three methods [1]. We show the results in Table 1.

We provide two variants our model in this experimental setup. The two models differ only in the size of the Weights Generator. For the "small" model the Weights Generator consist of $B = 1$ self-attention blocks followed by a fully connected layer of 40 hidden units. The self-attention block consists of three parallel weight projections of 40 dimensions followed by fully connected layers of 80 and 40 hidden units respectively. The "large" model consists

Table 2: Mean-Squared Error results for the alternative Sinusoidal Regression Task. Lower is better.

| Method | Alt. Sinusoidal 1000 tasks | |
|---|---|---|
| | 10 shot | 5 shot |
| EMAML | $1.524 \pm 0.034$ | $2.238 \pm 0.045$ |
| BMAML | $1.412 \pm 0.033$ | $2.157 \pm 0.049$ |
| Ours | $0.918 \pm 0.051$ | $2.389 \pm 0.103$ |
| Ours(Ensemble) | $\mathbf{0.630 \pm 0.035}$ | $\mathbf{1.857 \pm 0.081}$ |

[1]https://github.com/deepmind/neural-processes

of $B = 3$ self-attention blocks also followed by a fully connected layer of 40 hidden units. Each self-attention block has weight projections of 64 dimensions followed by fully connected layers of 128 and 64 hidden units respectively. Both MAML and Meta-SGD uses an architecture of 2 fully connected layers with 40 hidden units which is similar to the architecture of the Basis Learner network, though both Meta-SGD and MAML both have additional optimization for individual tasks. The Neural Process family of methods uses encoder archtecture of 4 fully connected layers with 128 hidden units and decoder architecture of 2 fully connected layers of 128 hidden units respectively which is more similar in architecture our larger model.

Similarly, we also compare our methods against two variants of EMAML and BMAML. The "small" model consist of 2 fully connected layers with 40 hidden units each while the "large" model consists of 5 fully connected layers with 40 hidden units each. This is to ensure fair comparison as both BMAML and EMAML lack a separate network to generate weight vectors but are ensemble methods that aggregate results from $M_p$ number of model instances. We set the number of model instances in BMAML and EMAML to 10.

**Alternative Sinusoidal Regression** We also evaluate our method on another version of the sinusoidal task as introduced by Yoon et al. (2018). The range of $A$ remain the same while the range of $b$ is increased to $[0, 2\pi]$ and the range of $\omega$ is increased to $[0.5, 2.0]$. An extra noise term, $\epsilon$ is also added the function $y(x)$. For noise $\epsilon$, we sample it from distribution $N \sim (0, (0.01A)^2)$. We also fix the total number of our tasks used during training to 1000 as in (Yoon et al., 2018). For this experimental setup we also include an ensemble version of our model where we train 10 separate instance of our model on the same 1000 tasks and aggregate their results by taking a mean of the predictions. We evaluate our model for both 10 shot and 5 shot cases and show the mean-squared error results in Table 2. For this experimental setup, we calculate the mean-squared error from 10 randomly points from 1000 advanced sinusoidal tasks.

Our results show that our method outperforms all recent few-shot regression methods in sinusoidal tasks.

## 4.2 2D REGRESSION ON IMAGE DATA

We also tested our method on more challenging image data, as done in (Garnelo et al., 2018a;b; Kim et al., 2019). We use MNIST (LeCun et al., 1998) and CelebA datasets (Liu et al., 2015) here for qualitative and quantitative comparison. Each image can be regarded as a continuous function $f : \mathbb{R}^2 \to \mathbb{R}^{d_y}$, where $d_y = 1$ if the image is gray-scale or or $d_y = 3$ if it is RGB. The input $\boldsymbol{x} \in \mathbb{R}^2$ to $f$ is the normalized coordinates of pixels and the output $\boldsymbol{y} \in \mathbb{R}^{d_y}$ is the normalized pixel value. The size of the images is $28 \times 28$ in MNIST and rescaled to $32 \times 32$ in CelebA. During meta-training, we randomly sample $K$ points from 784(1024) pixels in one image as $\mathbb{D}_{train}$ and another $K$ points as $\mathbb{D}_{val}$ to form a regression task. In the meta-testing stage, the MSE is evaluated on $784(1024) - K$ pixels. 60,000(162,770) images are used for meta-training and 10,000 for meta-testing for MNIST(CelebA) dataset.

We compare our methods with NP family: CNP (Garnelo et al., 2018a), NP(Garnelo et al., 2018b) and ANP (Kim et al., 2019) for $K = 50$ and $K = 100$. Deeper network structure is adopted due to the complexity of regression on image data. Namely, we use 5 fully connected layers with 128 hidden units in Basis Function Learner and 3 attention blocks in Weight Generator for our method. The encoders and decoders in NP family are all MLPs including 4 fully connected layers with 128 hidden units. Thus, the comparison is fair in terms of network capacity. All the models are trained for 500 epochs with batch size 80. The MSE on 10,000 tasks from meta-testing set is reported with $95\%$ confidence interval, shown in Table 3. The top results are highlighted. It can be observed that our method outperforms two of three NP methods and achieves MSE very close to most recent ANP. The outputs of regression on CelebA image data are high-dimension predictions, which demonstrates the effectiveness of our method in such challenging tasks. *Note that ANP significantly improves upon NP and CNP using cross-attention, which can potentially be applied to our method as well.*

Figure 2 shows the qualitative results for testing images. Red pixels in the images denote points in $\mathbb{D}_{train}$. The comparison with methods from NP family is shown in the Figure 2a. The regression by our method is clearly better than NP and CNP in 50-shot and 100-shot, which visually validates the quantitative results above. The qualitative results on CelebA can be found in Figure 7 in Appendix.

Table 3: Mean Square Error ($\times 10^{-2}$) for 2D regression on image data.

| Method | MNIST | | CelebA | |
|---|---|---|---|---|
| | 50-shot | 100-shot | 50-shot | 100-shot |
| CNP (Garnelo et al., 2018a) | 4.13±0.033 | 3.23±0.024 | 2.74±0.023 | 2.37±0.019 |
| NP (Garnelo et al., 2018b) | 4.14±0.030 | 3.26±0.024 | 2.51±0.019 | 2.15±0.017 |
| ANP (Kim et al., 2019) | **3.40±0.026** | **2.20±0.017** | **2.13 ±0.018** | **1.63±0.015** |
| Ours | **3.65±0.028** | **2.43±0.015** | **2.39±0.021** | **1.93±0.015** |

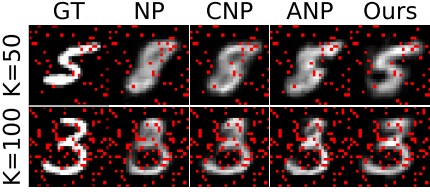

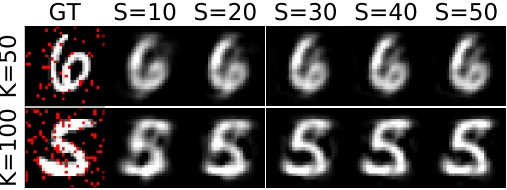

(a) Comparison with NP methods.  (b) Prediction with the first **S** largest weight parameters.

Figure 2: Qualitative results on MNIST image data.

## 4.3 ANALYSIS ON BASIS FUNCTIONS

In this subsection we provide some deeper analysis on the basis functions that are learned by our method. In particular, we provide some further evidence to our claim that our method learns a set of sparsifying basis functions that correspond to the regression tasks that we would like to model. To demonstrate the sparsity of basis functions, we take **only** the $S$ largest weights in terms of $|w|$ and their corresponding basis functions and illustrate the predicted regression function with the combination of **only** the $S$ weights and basis functions. We conduct this experiment on both the sinusoidal regression task and the more difficult image completion task and show these $S$-weights predictions in Figures 3 and 2b respectively.

The figures illustrate that our method is able to produce a good prediction of the regression function with only a fraction of the full set learned basis function (40 for the sinusoidal task, 128 for the MNIST image completion task). This demonstrates the sparsity of $\Phi(\boldsymbol{x})$ as most of the prediction is carried out by just a small number of basis functions. This also demonstrates that our method is able to force most of the information of $F(\boldsymbol{x})$ to be contained in a few terms. Therefore, using $K$ samples to estimate the weights of these few important terms could achieve a good approximation of $F(\boldsymbol{x})$.

## 4.4 ABLATION STUDIES

In this subsection we detail some ablation studies on our model to test the validity of certain design choices of our model. In particular we focus on the effects of the addition of self-attention operations in the Weights Generator and also the effects of using different penalty terms on our loss function.

To test out **the effects of adding the self-attention operations** to our model, we conduct a simple experiment where we replace the self attention operations in the self-attention block with just a single fully connected layer of equal dimensions as the self-attention weight projection. Essentially, this reduces the Weights Generator to be just a series of fully connected layers

Table 4: Comparison of Models with and without Self-Attention on 10-shot Sinusoidal Regression

| | 10-shot |
|---|---|
| Without Attention | $0.163 \pm 0.003$ |
| With Attention | $\mathbf{0.062 \pm 0.003}$ |

with residual connections and layer normalization. We compare the simpler model performance on the sinusoidal regression task as specified in Table 1 with our original model and show the results in Table 4. The results show that adding the self-attention operations do improve our methods performance on the 1D sinusoidal regression task.

We also conducted experiments to test **the effects of the different penalty terms** on the the generated weights vector. In this ablation study, we compared our models trained using different variants of the

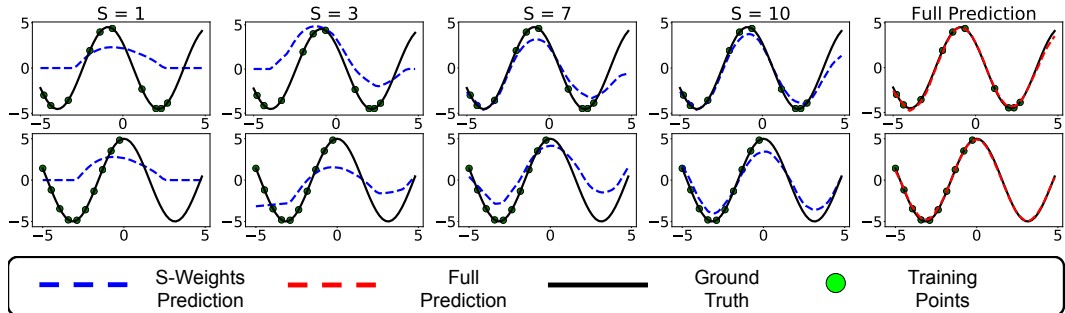

Figure 3: Top S-weights predictions for sinusoidal tasks

loss function we presented in Equation 5. Similar to the previous study, we evaluate them on their performance on the sinusoidal regression task as specified in Table 1. The variants we tested out are: (i) Loss function with only the L1-norm penalty term ; (ii) Loss function with only the L2-norm penalty term (iii) Loss function with both L1 and L2-norm penalty terms. To demonstrate the sparsity of the weights vectors of each variant, we also show the a breakdown of the magnitude of the learned weight vectors over 100 sinusoidal tasks. We group the weight vectors into three groups : $|w|$ less than 0.02 to indicate weights that are near zero, $|w|$ between 0.02 and 1 and weights with magnitude more than 1. We show the results of the different variants in Table 5. We also present histograms of the magnitude of the learned weight vectors in Figure 4

The results do show that the combination of both L1 and L2 penalty terms do ultimately give the best performance for the sinusoidal regression task. In terms of sparsity, the model trained with only the L1 loss term do gives the highest percentage of sparse weights though we found the model with both L1 and L2 terms do give a better performance while still maintaining a relatively high percentage of near zero weights.

## 5 CONCLUSION

We propose a few-shot meta learning system that focuses exclusively on regression tasks. Our model is based on the idea of linear representation of basis functions. We design a Basis Function Learner network to encode the basis functions for the entire task distribution. We also design a Weight generator network to generate the weights from the $K$ training samples of a novel task drawn from the same task distribution. We show that our model has competitive performance in in various few short regression tasks.

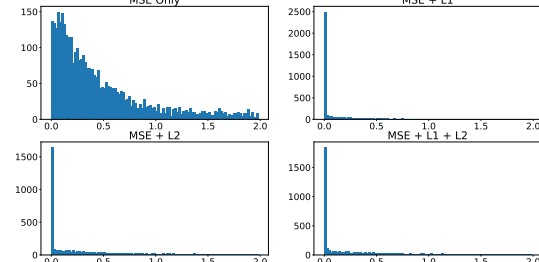

Figure 4: Histogram of magnitude of learned weight over 100 Sinusoidal tasks.

Table 5: Comparison of models trained using different penalty for 10-shot Sinusoidal Regression.

| Penalty | 10-shot | Percentage of Weight, $w$ by Magnitude | | |
| | | $|w| < 0.02$ | $0.02 <= |w| < 1.0$ | $|w| >= 1.0$ |
| --- | --- | --- | --- | --- |
| L1 only | $0.083 \pm 0.004$ | 62.45 | 32.925 | 4.625 |
| L2 only | $0.073 \pm 0.004$ | 41.1 | 43.575 | 15.325 |
| L1 + L2 | $\mathbf{0.062 \pm 0.003}$ | 46.1 | 41.75 | 12.15 |

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

## A    VISUALIZATION OF SINUSOIDAL BASIS FUNCTIONS

Figure 5: Illustration of all the non-zero basis functions that are learnt by our method on the sinusoidal regression task by order of magnitude.

In this section we illustrate all of the individual non-zero basis functions learned by our model for the sinusoidal task. These functions are shown Figure 5. Note that out of 40 of the basis functions, only 22 of the learned basis functions are non-zero functions, further demonstrating that indeed our method is forcing the model to learn a set of sparse functions to represent the tasks. Furthermore, it can be seen that the learned basis functions all correspond to the different "components" of sinusoidal function: most of the learned functions seem to represent possible peaks, or troughs if multiplied with a negative weight at various regions of the input range whereas the top four basis function seem to model the more complicated periodic nature of the sinusoidal functions.

## B    ADDITIONAL ANALYSIS ON LEANED BASIS FUNCTIONS

Adding on to the experiments in Section 4.3, we also illustrate what happens when do the exact opposite. We take the prediction using the full set of weight vectors/basis function and study the effect of the prediction when we remove certain basis function from the prediction. Similar to the previous experiment, we remove the basis function by order of magnitude starting with the basis function with the largest corresponding $|w|$. we conduct this experiment on the sinusoidal regression task and illustrate the results in Figure 6.

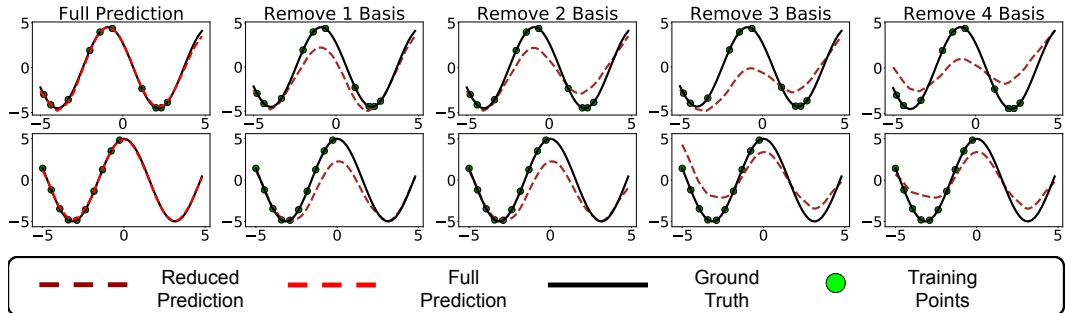

Figure 6: Reduced predictions for sinusoidal tasks

Similarly, this study also demonstrates the importance of certain basis functions as removing them caused the prediction the change drastically. In particular, notice that for sinusoidal task, removing just 4 of the most important basis functions resulted in a less accurate prediction than using just 10 of the most important basis functions.

## C    DETAILS ON THE WEIGHTS GENERATOR NETWORK ARCHITECTURE

Here we provide more details on the architecture of the Weights Generator Network. As mentioned previously in Section 3.3. The Weights Generator Network consists of a series of self attention blocks followed by a final fully connected layer. We define a self attention block as such: An attention block consists of a self attention operation on the input of self attention block. Following the self-attention operation, the resultant embedding is further passed through two fully connected layers. A residual connection (He et al., 2016) from the output of the self-attention operation to the output of the second fully connected layer. Finally, resultant embedding of the residual connection is then passed though a a layer normalization operation (Ba et al., 2016). Note that the input of the first self attention block will always be the input to the Weights Generator network, $(\Phi(\boldsymbol{x}_t^k), \boldsymbol{y}_t^k)$ whereas the inputs to subsequent attention blocks are the outputs of the previous attention block.

For the self-attention operation, the input is transformed into query, key and value vectors though their respective weight projections. These query, key and value vectors, $Q$, $K$ and $V$ then go through a scaled dot-product self-attention operation as introduced by (Vaswani et al., 2017):

$$Att(Q, K, V) = softmax(\frac{QK^T}{\sqrt{d_k}})V, \qquad (6)$$

## D    ADDITIONAL REGRESSION EXPERIMENTS

**1D Heat Equation**.  We also evaluate our method on another 1D Regression task, the 1D heat Equation task, we define it as such: Consider a 1-dimensional rod of length $L$ with both of its ends connected to heat sinks, *i.e.* the temperature of the ends will always be fixed at $0K$ unless a heat source is applied at the end. a constant point heat source is then applied to a random point $s$ on the rod such the the heat point source will always have a temperature of $1.0K$. We would like the model the temperature $u(x, t)$ at each point of the rod a certain time $t$ after applying the heat source until the temperature achieves equilibrium throughout the rod. The temperature at each point $x$ after time $t$ is given by the heat equation:

$$\frac{\partial u}{\partial t} = k \frac{\partial^2 u}{\partial x^2}$$

For our experiments, we set $L$ to be 5 and randomly sample K points of range $[0, 5]$ on the heat equation curve. We fix the total number of tasks used during training to 1000 and evaluate our model on both 10 shot and 5 shot cases, similar to the experimental setup for the Advanced Sinusoidal tasks. We also compare our results to both EMAML and BMAML on this regrssion task and add an ensemble version of method for comparison.The results of our evaluation is presented in Table 6.

**2D Gaussian**. We also evaluated our method on the for the 2D Gaussian regression tasks. For this task, we train our model to predict the probability distribution function of a two-dimensional Gaussian

Table 6: Mean-Squared Error results for the 1D Heat Equation Regression Task.

| Method | 1D Heat Equation 1000 tasks | |
| --- | --- | --- |
| | 10 shot | 5 shot |
| **EMAML (Finn et al., 2017; Yoon et al., 2018)** | $(1.02 \pm 0.03) \times 10^{-2}$ | $(1.49 \pm 0.03) \times 10^{-2}$ |
| **BMAML (Yoon et al., 2018)** | $(1.74 \pm 0.04) \times 10^{-2}$ | $(1.81 \pm 0.04) \times 10^{-2}$ |
| **Ours** | $(6.32 \pm 0.53) \times 10^{-3}$ | $(8.30 \pm 0.56) \times 10^{-3}$ |
| **Ours(Ensemble)** | $\mathbf{(5.13 \pm 0.13) \times 10^{-3}}$ | $\mathbf{(7.09 \pm 0.16) \times 10^{-3}}$ |

Table 7: Mean-Squared Error results for the 2D Gaussian Regression Task.

| Method | 2D Gaussian 1000 tasks | | |
| --- | --- | --- | --- |
| | 10 shot | 20 shot | 50 shot |
| **EMAML** | $(2.67 \pm 0.17) \times 10^{-3}$ | $(2.44 \pm 0.13) \times 10^{-3}$ | $(2.16 \pm 0.89) \times 10^{-3}$ |
| **BMAML** | $(2.26 \pm 0.15) \times 10^{-3}$ | $(2.16 \pm 0.09) \times 10^{-3}$ | $(1.48 \pm 0.07) \times 10^{-3}$ |
| **Ours** | $(1.70 \pm 0.36) \times 10^{-3}$ | $(1.14 \pm 0.21) \times 10^{-3}$ | $(7.83 \pm 0.96) \times 10^{-4}$ |
| **Ours(Ensemble)** | $\mathbf{(1.46 \pm 0.11) \times 10^{-3}}$ | $\mathbf{(0.97 \pm 0.10) \times 10^{-3}}$ | $\mathbf{(6.09 \pm 0.67) \times 10^{-4}}$ |

distribution. We train our model from Gaussian distribution task with mean ranging from $(-2, -2)$ to $(2, 2)$ and standard deviation of range $[0.1, 2]$. We fix the standard deviation to be of the same value in both directions. Similar to the heat equation, we use the same setup as the Advanced Sinusoidal task and compare our methods to EMAML and BMAML. We evaluate our model on 10, 20 and 50 shot case. The results of our evaluation is presented in Table 7.

**Qualitative results on CelebA datasets.** We provide the qualitative results on CelebA datasets in Figure 7. We note that the RGB images are complex 2D functions. We choose them to evaluate so that we can see the results more directly, not to compare with image inpainting methods, which is also mentioned in (Garnelo et al., 2018a). The results in Figure 7a are consistent with Figure 2a. The regression results from our method are visually better than NP and CNP. The predictions using first $S$ largest weights are shown in Figure 7b. The 2D image function is usually predicted with less than 50 weights, which suggests that the information of the 2D function is kept in several terms.

# E    COMPARISON WITH DICTIONARY LEARNING

Our proposed sparse linear representation framework for few shot regression makes the few shot regression problem appears to be similar to another research problem called dictionary learning (DL), which focuses on learning dictionaries of atoms that provide efficient representations of some class of signals (Tosic & Frossard, 2011). However the differences between DL and our problem are significant: Our problems are continuous rather than discrete as in DL, and we only observe a very small percentage of samples.

Specifically, for a given $y \in R^n$, the goal of DL is to learn the dictionary ($n$ by $M$) $\Phi$ for some sparse $w$:

$$y = \Phi w \tag{7}$$

In typical DL, the *entire* $y$ is given. Also, $M > n$ for an overcomplete dictionary (Figure 8).

In few shot regression, the goal is to predict the entire continuous function $y = F(x)$. Therefore, viewing this as the setup in (7), $n$ is infinite. Moreover, only a few ($K$) samples of $y$ is given: $y_t^k = F(x_t^k)$. The locations of the given samples ($x_t^k$) are different for different $y$ (different task). Therefore, our problem is significantly different and more difficult than DL. Typical DL algorithms solve (7) and return $\Phi$, which is a $n$ by $M$ matrix of finite dimensions (the dictionary). In our setup, the basis matrix $\Phi$ has infinite entries, and $\Phi$ is encoded by the proposed Basis Function Learner network.

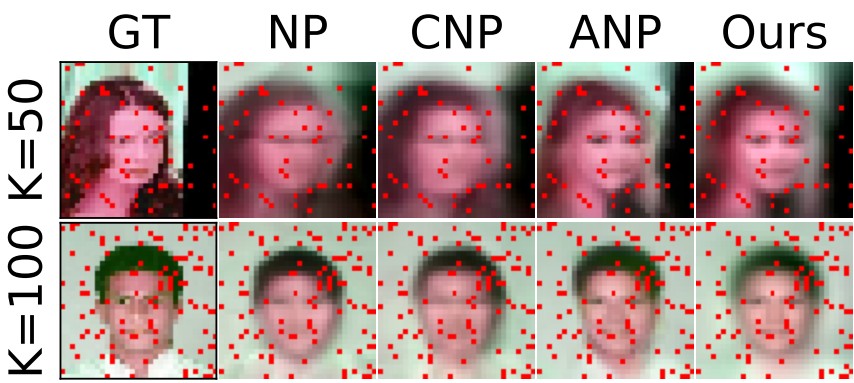

(a) Comparison with NP methods.

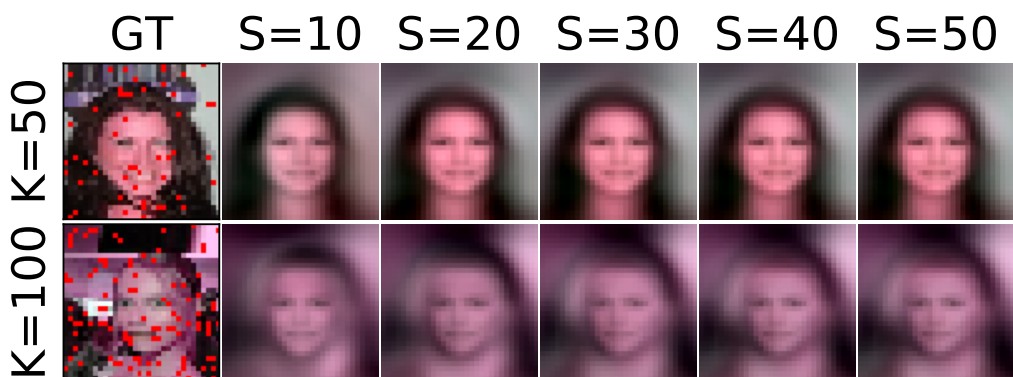

(b) Prediction with only the basis functions with the **S** largest weights.

Figure 7: Qualitative results on CelebA image data.

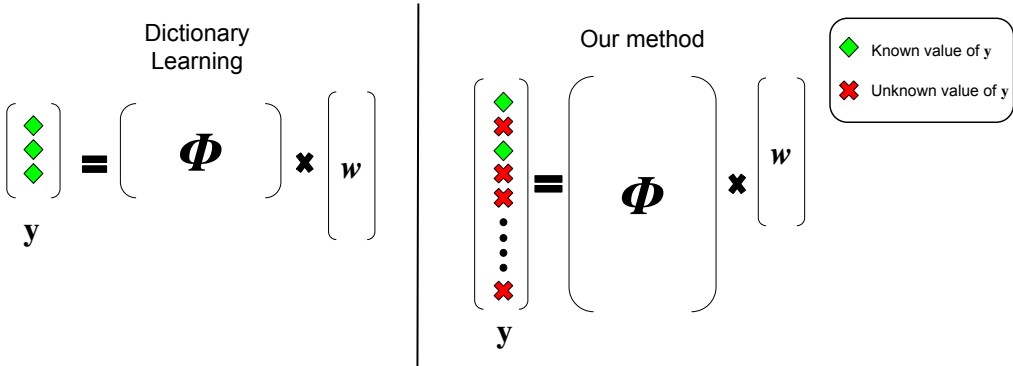

Figure 8: Comparing our method to Dictionary Learning

