# OpenReview forum: "Few-Shot Regression via Learning Sparsifying Basis Functions"
_ICLR.cc/2020/Conference — Reject_

### Official Review · AnonReviewer2 · 2019-10-22
**Official Blind Review #2**

**Rating:** 3

**Review:**

The paper proposes a regression approach that, given a few training (support) samples of a regression task (input and desired output pairs), should be able to output the values of the target function on additional (query) inputs. The proposed method is to learn a set of basis functions (MLPs) and a weight generator that for a given support set predicts weights using which the basis functions are linearly combined to form the predicted regression function, which is later tested (using the MSE metric) w.r.t. the ground truth. The method is trained on a large collection of randomly sampled task from the target family and is tested on a separate set of random tasks. The experiments include:
* sinusoidal wave prediction from a few samples
* MNIST and CelebA inpainting from a set of known pixel values (in 28x28 and 32x32 resolution respectively)
* additional experiments on heat equation and 2D Gaussian distribution task in Appendix
The experiments show that the proposed approach outperforms the other methods on the sinusoidal wave toy problem, and yet performs less good then than Kim et al. 2019 on MNIST and CelebA.

I propose to reject the paper in its current form, and consider the following negative points for further improvement:

1. The posed problem is not really few-shot learning, in (now classical) few-shot learning, such as few-shot classification on benchmarks such as miniImageNet, CUB, tieredImageNet, CIFAR-FS, FC100, etc. the meta-training is done on a disjoint set of categories and testing is done on a completely new set of categories unseen during training. The gap between disjoint visual categories is very large, and it does not come close to being tested on a different from training samples sinusoidal wave or different set of hidden pixels in inpainting on the same (seen during training) set of classes (where the basis function module could learn a set of basis functions for every class). In the proposed setting, I think a better definition would be "learning a structured regression" from a set of sample points to a function, and not few-shot regression.
If the authors would like to keep the "few-shot flavour", I would suggest re-formulating the experiments, and meta-train on some set of classes (e.g. inpainting over digits 0 to 4) and meta-test on a different set of classes (e.g. inpainting over digits 5 to 9). This partially holds for faces as they are all mostly different categories (different people), but in 32x32 resolution and MSE metric, I don't think they are sufficiently different.

2. I would expect stronger results on the more realistic MNIST and CelebA experiments (although as suggested in 1. the setting there should be different), currently it does less well then existing method.

3. An emerging important class of few-shot regression problems is few-shot object detection, where the bounding box coordinates of objects location need to be regressed. There are several papers and benchmarks in this space, and it will help the current paper to test on this challenging family of problems. Please see the following papers for benchmarks and settings:
* LSTD: A Low-Shot Transfer Detector for Object Detection, Chen et al. 2018
* RepMet: Representative-based metric learning for classification and one-shot object detection, Karlinsky et al. 2019
* Few-shot Object Detection via Feature Reweighting, Kang et al. 2019

**Experience Assessment:**

I have published one or two papers in this area.

**Review Assessment: Checking Correctness Of Derivations And Theory:**

I assessed the sensibility of the derivations and theory.

**Review Assessment: Checking Correctness Of Experiments:**

I carefully checked the experiments.

**Review Assessment: Thoroughness In Paper Reading:**

I read the paper thoroughly.

---

> ### Author Response · Authors · 2019-11-15
> **Reply to Reviewer 2**
>
> Thank you for a thorough reading and review of our paper. We will try our best to address your comments/concerns below.
>
> We thank you for the suggestion of an alternative experimental setting of the few-shot image completion task with disjoint classes in training and testing. We will include this experimental setup in future versions of the paper.
>
> We will strive to include experiments of more realistic regression tasks in the future versions of the paper and we thank you for giving us a list of related works that we could look into and compare against.

---

### Official Review · AnonReviewer1 · 2019-10-23
**Official Blind Review #1**

**Rating:** 3

**Review:**

The authors propose using sparse adaptive basis function models for few shot regression. The basis functions and the corresponding weights are generated via respective networks whose parameters are shared across all tasks. Elastic net regularization is used to encourage task specific sparsity in the weights,  the idea being  that with only a small number of available training examples, learning a sparse basis is  easier than learning a dense basis with many more parameters. The method is validated on both synthetic data and on image completion tasks.

I am leaning towards rejecting the paper. 1) Although, the paper is well written and easy to follow the technical contributions of the paper are limited. Adaptive basis functions and their sparse combinations are decades old ideas.  While the application of these ideas to few shot regression does appear to be novel,  this combination don’t seem to provide an obvious improvement over existing alternatives. 2) The empirical evidence presented is rather limited, the proposed approach only seems to outperform competitors on the synthetic sinusoidal regression experiments. Lack of strong empirical performance along with the limited novelty

Detailed comments and questions:
+ The approach naturally extends to few shot classification problems once the MSE loss in Equation 5 is replaced with an appropriate cross entropy loss. Was this considered and is the approach competitive on few shot classification problems.

+ The empirical section could be significantly improved.

 -  Diverse synthetic data: I don’t see the value in presenting two sets of synthetic sinusoid regression experiments.  It would be better to replace the alternative sinusoid task with qualitatively different tasks. This would help the  audience ascertain whether the favorable performance demonstrated in Table 1 generalizes beyond sinusoidal signals.

 - Comparisons:  1.  Why are comparisons to neural processes missing in the additional synthetic experiments presented in the supplement and from Table 2? This is a particularly egregious omission since on the real data (attentive) neural processes outperform the proposed method.
2. The ensemble approach seems to improve on the individual model significantly in Table 2. Why was this not considered for the image completion experiments? The authors would also do well to more clearly describe how the ensembling was performed.

+ I find it curious that the basis functions are restricted to be non-negative. The description in 3.3 suggests that the basis function network outputs are passed through a ReLU.  What was the rational behind this design choice?

Minor:
 Why are both ANP and “Ours” highlighted in Table 3, when ANP clearly outperforms and does not appear to be within statistical noise of “Ours”.

**Experience Assessment:**

I have read many papers in this area.

**Review Assessment: Checking Correctness Of Derivations And Theory:**

N/A

**Review Assessment: Checking Correctness Of Experiments:**

I assessed the sensibility of the experiments.

**Review Assessment: Thoroughness In Paper Reading:**

I read the paper thoroughly.

---

> ### Author Response · Authors · 2019-11-15
> **Reply to Reviewer 1**
>
> Thank you for a thorough reading and review of our paper. We will try our best to address your comments/concerns below.
>
> It is true our method might be able to be extended to classical few-shot classification tasks. However the main idea of our paper is to learn an optimal combination basis that can be used to predict a regression function, we choose to limit our experiments and evaluations to just few-shot regression tasks.
>
> Regarding your comment on more diverse/realistic datasets, we will strive to include experiments with more realistic regression tasks in future versions/submissions of the paper.
>
> For comparisons of ANP/NP/CNP against our method, we show the results below:
>
>                                           |            ANP                    |              NP                    |             CNP                 |
> --------------------------------------------------------------------------------------------------------------------------------
> Alt. Sinusoidal 10 shot   |  1.234 +- 0.075             |      3.240 +- 0.125         |     3.045 +- 0.120        |
> Alt Sinusoidal 5 shot      |  2.613 +- 0.109             |      3.829 +- 0.125         |     3.686 +- 0.125        |
> --------------------------------------------------------------------------------------------------------------------------------
> 1D Heat Eqn 10 shot      | (6.02 +- 0.50)*10^-3    | (1.18 +- 0.06)*10^-2   | (1.04 +- 0.06)*10^-2 |
> 1D Heat Eqn 5 shot        | (7.88 +- 0.47)*10^-3    | (1.41 +- 0.07)*10^-2   | (1.42 +- 0.08)*10^-2 |
> --------------------------------------------------------------------------------------------------------------------------------
> 2D Gaussian 10 shot      |  (1.26 +- 0.26)*10^-3   |  (1.40 +- 0.29)*10^-3  | (1.30 +- 0.28)*10^-3 |
> 2D Gaussian 20 shot      |  (5.67 +- 1.10)*10^-4   |  (7.05 +- 1.65)*10^-4  | (7.06 +- 1.45)*10^-4 |
> 2D Gaussian 50 shot      |  (2.38 +- 1.22)*10^-4   |  (4.51 +- 0.85)*10^-4  | (4.39 +- 0.96)*10^-4 |
> --------------------------------------------------------------------------------------------------------------------------------
>
> We note that our method outperforms the NP family of methods for the alternative sinusoidal task but is slightly worse in performance compared to ANP for the Heat Equation task and is worse than all NP methods for the Gaussian task. The gap in performance on the Gaussian task certainly indicates that there is room for improvement in our method and we will take that into account in our future submissions.
>
> The Ensemble results of our method, as specified in Section 4.1 consist of 10 separately instances of our model (with randomly initialized weights) trained on the same set of regression tasks. The final prediction of the ensemble model is obtained by taking a mean of predictions of the 10 separate models. As for the ensemble results for the image completion task, we found that the ensemble version of our method does perform slightly better than ANP in the MNIST image completion task (2.12e-2 for 100-shot). Though we note It is not an equivalent comparison against the NP methods as they themselves are not ensemble methods.
>
> You are correct to note that the outputs of the Basis Function Learner are passed through a ReLU activation function. We choose this particular design choose to emulate the structure of traditional neural networks as the output of the Basis Function Learner can be seen as the penultimate layer of a neural network whereas the linear combination of the weights vector and the learned basis functions can be seen as the final layer of the network.

---

### Official Review · AnonReviewer3 · 2019-10-30
**Official Blind Review #3**

**Rating:** 3

**Review:**

This paper describes an approach to few-shot regression based on
learning a sparse basis and a weight estimator network. The authors
introduce a sparsity inducing term in the loss to encourage sparse
weight generation for tasks; these sparse coefficient vectors are then
projected onto the learned, task-dependent basis for
regression. Experimental results are given on two synthetic regression
problems, with a comparison with the recent state-of-the-art.

This paper has some interesting ideas in it. However, it does have
some issues:

 1. Clarity. There are several points of the proposed technique that
    are not described clearly enough. For example, the diagram in
    Figure 1 leads me to believe that the basis and weights generators
    are independent (and thus not trained end-to-end). However, the
    loss in eq. (5) seems (though it is not completely clear to me) to
    depend on both networks (which is how I would expect things to
    work). Also, the "self attention blocks" mentioned at several
    points are never completely defines. And from the ablation study
    is seems that the improvement form self-attention is the lion's
    share of the overall improvement. I do not feel that it would be
    easy to reproduce the results reported in this paper without
    significant guesswork.

 2. The experimental results are somewhat limited. The sinusoidal
    regression problem is very artificial, as is the image regression
    task. Focusing on regression is inherently limiting, but results
    on more realistic regression problems would help establish more
    clearly the significance of the contribution.



**Experience Assessment:**

I have read many papers in this area.

**Review Assessment: Checking Correctness Of Derivations And Theory:**

I assessed the sensibility of the derivations and theory.

**Review Assessment: Checking Correctness Of Experiments:**

I assessed the sensibility of the experiments.

**Review Assessment: Thoroughness In Paper Reading:**

I made a quick assessment of this paper.

---

> ### Author Response · Authors · 2019-11-15
> **Reply to Reviewer 3**
>
> Thank you for your review and comments. We will do our best to address your comments/questions below.
>
> We apologize if our method is not clearly explained enough. Yes indeed as you pointed in Eq (5). Both the weights of the Basis Function Learner, \theta and Weights Generator \psi are optimized jointly end-to-end. A description of the self-attention block was included in the supplementary section C of the paper.
>
> As noted by you and other reviewers, we will include experiments with more realistic regression tasks in future versions/submissions of the paper.

---

### Public Comment · ~Anthony_Wittmer1 · 2019-09-28
**No code in the repo of the provided github link**

Hi,

No code is present in the repo of the github link. It is not fair to provide a placeholder link for code submissions (which impact the review process) and submit code taking considerable buffer time after submission deadline.

---

> ### Author Response · Authors · 2019-09-28
> **Code uploaded**
>
> Hi,
>
> Thank you for the comment.
>
> The link to the Github repo has been updated with our code for the paper.

---

### Decision · Program_Chairs · 2019-12-19

**Decision:**

Reject

**Comment:**

All reviewers agree that this paper is not ready for publication.